# Inactivation of Genes Encoding MutL and MutS Proteins Influences Adhesion and Biofilm Formation by *Neisseria gonorrhoeae*

**DOI:** 10.3390/microorganisms7120647

**Published:** 2019-12-04

**Authors:** Jagoda Płaczkiewicz, Monika Adamczyk-Popławska, Robert Lasek, Pawel Bącal, Agnieszka Kwiatek

**Affiliations:** 1Institute of Microbiology, Faculty of Biology, University of Warsaw, Miecznikowa 1, 02-096 Warsaw, Poland; j.placzkiewicz@biol.uw.edu.pl (J.P.); poplawa@biol.uw.edu.pl (M.A.-P.); lasek@biol.uw.edu.pl (R.L.); 2Faculty of Chemistry, University of Warsaw, Pasteura 1, 02-093 Warsaw, Poland; bacal@chem.uw.edu.pl; 3Nalecz Institute of Biocybernetics and Biomedical Engineering, Polish Academy of Sciences, 02-109 Warsaw, Poland

**Keywords:** *Neisseria gonorrhoeae*, gene expression, biofilm, adhesion, MutL protein, MutS protein

## Abstract

*Neisseria gonorrhoeae* is an etiological agent of gonorrhea, which remains a global health problem. This bacterium possesses MutL and MutS DNA repair proteins encoded by *mutL* and *mutS* genes, whose inactivation causes a mutator phenotype. We have demonstrated the differential gene expression in *N. gonorrhoeae mutL* and *mutS* mutants using DNA microarrays. A subset of differentially expressed genes encodes proteins that can influence adhesion and biofilm formation. Compared to the wild-type strain, *N. gonorrhoeae mutL* and *mutS* mutants formed denser biofilms with increased biofilm-associated biomass on the abiotic surface. The *N. gonorrhoeae mutS::km*, but not the *mutL* mutant, was also more adherent and invasive to human epithelial cells. Further, during infection of epithelial cells with *N. gonorrhoeae mutS::km*, the expression of some bacterial genes encoding proteins that can influence gonococcal adhesion was changed compared with their expression in cells infected with the wild-type gonococcus, as well as of human genes’ encoding receptors utilized by *N. gonorrhoeae* (CD46, CEACAM 1, HSPG 2). Thus, deficiency in the *mutS* gene resulting in increased mutation frequency in singular organisms can be beneficial in populations because these mutants can be a source of features linked to microbial fitness.

## 1. Introduction

*Neisseria gonorrhoeae* (gonococcus) is a human-specific, gram-negative diplococcus that causes the sexually transmitted disease gonorrhea. Gonococcal infections remain a global health issue, as highlighted by the World Health Organization, which has recently classified *N. gonorrhoeae* as a “priority pathogen” due to spreading multidrug resistance and increasing infection incidence [1,2]. Additionally, gonococcal infection may increase the risk of human immunodeficiency virus transmission [3,4,5]. During infection, *N. gonorrhoeae* interacts with host epithelial cells using multistep adhesion as the key stage of infection, allowing biofilm formation, which is an important virulence factor [6,7].

DNA repair systems are crucial in all living organisms. MutL and MutS proteins are conserved DNA repair molecules present at all domains of life. MutL is a “molecular matchmaker”, which mediates the protein–protein interactions during mismatch recognition and strand removal, and MutS recognizes unpaired and mispaired bases and small insertion/deletion loops in duplex DNA [8]. In *N. gonorrhoeae*, inactivation of *mutL* and *mutS* genes leads to an overall increase in mutability. Thus far, it has been demonstrated that gonococcal MutL and MutS proteins remove mononucleotide mismatches and insertion-deletion loops, indicating their engagement in the methyl-directed mismatch repair system (MMR) and affect the activity of Vsr endonucleases, implying a role in the Very Short Patch repair system [9,10]. Furthermore, in addition to the engagement of MutL and MutS proteins in DNA repair systems, they limit the frequency of RecA-dependent gene conversion events at the *pilE* gene, and MutS protein prevents recombination between partially homologous sequences [9,11].

There are data indicating that besides maintaining the integrity of the genome, DNA repair systems can also play broader functions affecting the intraspecific variation of strains, host defense avoidance mechanisms, improved adaption to changing environmental conditions, and virulence [12,13,14]. In *Neisseria meningitidis*, in addition to repairing DNA damage [15], the MMR system plays an important role in infection, as it was demonstrated that epidemic *N. meningitidis* serogroup A isolates were found to be strains with an increased frequency of mutations caused by defects in the MMR [16]. Further, the meningococcal *mutS* mutant can be regarded as more pathogenic, because it has higher ability to escape from the bactericidal activity of monoclonal antibodies [17]. Whereas for *Escherichia coli*, *Salmonella* spp., or *Pseudomonas aeruginosa*, the occurrence of mutator strains in population was demonstrated [18,19,20].

To date, there are no data on the influence of disruption of *mutL* or *mutS* genes on processes other than DNA repair in *N. gonorrhoeae*. Therefore, the goal of this work was to evaluate the influence of the disruption of *mutL* and *mutS* genes on processes connected with gonococcal infection, specifically on biofilm formation and interactions with human cells, detailed on adhesion and invasiveness to epithelial cells. Our results indicated that *N. gonorrhoeae* with a disrupted *mutS* gene formed denser biofilms and was more adherent and invasive to human epithelial cells compared to the wild-type strain.

## 2. Materials and Methods

### 2.1. Bacterial Strains and Growth Conditions

*N. gonorrhoeae* FA1090 and its derivatives were used in the study. *N. gonorrhoeae* FA1090 (ATCC^®^ 700825) was originally isolated from the endocervix of a woman with a probable disseminated gonococcal infection in 1981 [21]. The strain is well-characterized, serum-resistant, streptomycin-resistant, and proline-requiring. *N. gonorrhoeae* strain FA1090 is extensively used in studies of gonococcal pathogenesis, as it is the first gonococcal strain for which the entire genome sequence was determined (NC_002946.2).

*N. gonorrhoeae* FA1090 and its derivatives were grown at 37 °C on a GC agar base (Difco, Detroit, MI, USA) supplemented with 1% Kellogg’s supplement in 5% CO_2_, and depending on the experiment, with or without 1% hemoglobin or cultivated in GC broth supplemented with 1% Kellogg’s supplement and 0.043% NaHCO_3_ [22]. When needed, media were supplemented with 30 μg/mL kanamycin or/and 2 μg/mL chloramphenicol.

For all experiments, gonococci exhibiting the same piliation and opaque phenotype were used as was determined by microscopy using stereo dissecting microscope (PZO Mikroskopy, Warsaw, Poland) observation in accordance with the principles described by Dillard [22].

Each set of experiments has always included: *N. gonorrhoeae* wild-type, *N. gonorrhoeae mutL::km*, and *N. gonorrhoeae mutS::km* mutants, complementation gonococci: *N. gonorrhoeae mutL::km* + *mutL_wt_* and *N. gonorrhoeae mutS::km* + *mutS_wt_*.

### 2.2. Cell Lines and Culture Conditions

The human epithelial cell line HEC-1-B is one of the models for infection of human tissue culture cells by *N. gonorrhoeae* [23]. The use of HEC-1-B cells for the study of gonococcal infection of epithelial cells has already been described [24,25,26,27,28]. The epithelial cell line HEC-1-B (ATCC: HTB113) was grown in DMEM medium (Biowest, Nuaillé, France) supplemented with 10% fetal bovine serum (FBS) (Biowest) (*v*/*v*), 2 mM l-glutamine, 1 mM sodium pyruvate and, when necessary, with 100 µg/mL penicillin/streptomycin (Biological Industries, Cromwell, CT, USA) or 2.5 µg/mL fungizone (Biological Industries) at 37 °C and 5% CO_2_. Confluence was determined by visual examination.

### 2.3. Construction of Neisseria gonorrhoeae (N. gonorrhoeae) Mutants

Construction of *N. gonorrhoeae mutL::km* and *N. gonorrhoeae mutS::km* mutants along with the obtained complementation strains *N. gonorrhoeae mutL::km* + *mutL_wt_* and *N. gonorrhoeae mutS::km* + *mutS_wt_*, has been previously detailed described in Adamczyk-Popławska et al. [10].

### 2.4. Field Emission Scanning Electron Microscopy (FE SEM)

Field emission scanning electron microscopy (FE SEM) (Carl Zeiss, Oberkochen, Germany) was performed as described previously with slight modifications [27,28,29]. First, bacterial suspensions (10^7^ cells/mL) were made (in GC broth to study biofilm on abiotic surface or in DMEM to visualization of *N. gonorrhoeae* attached to human epithelial cells) and then were added: (i) to coverslips placed in Petri dishes, and bacteria were cultivated for 24 h at 37 °C in 5% CO_2_ for study of *N. gonorrhoeae* biofilm formation on abiotic surface, or (ii) to human epithelial cells (m.o.i 1:100) seeded on coverslips for 4 h at 37 °C in 5% CO_2_ for study of *N. gonorrhoeae* attached to epithelial cells. Visualization of biofilms formed on abiotic surface or of *N. gonorrhoeae* attached to epithelial cells was performed as described in Kwiatek et al. [27,28].

### 2.5. Scanning Confocal Laser Microscopy (SCLM)

Scanning confocal laser microscopy (SCLM) was performed as described previously with slight modifications [27,28,29]. First, bacterial suspensions (10^7^ cells/mL) were made (in GC broth to study biofilm on abiotic surface or in DMEM to visualization of *N. gonorrhoeae* attached to human epithelial cells). Then: (i) bacterial suspensions were added to glass bottom microwell dishes (MatTek Corporation, Ashland, MA, USA) and bacterial growth was continued at 37 °C in 5% CO_2_ for 24 h for study of *N. gonorrhoeae* biofilm formation on abiotic surface, or (ii) to human epithelial cells (m.o.i 1:100) seeded on coverslips for 4 h at 37 °C in 5% CO_2_ for study of *N. gonorrhoeae* attached to epithelial cells. Next, *N. gonorrhoeae* biofilm formed on abiotic surface was washed with 10 mM MgSO_4_, stained for 30 min with acridine orange solution (10 µg/mL in 10 mM MgSO_4_) and rinsed with 10 mM MgSO_4_. Whereas, to investigate *N*. *gonorrhoeae* attached to human cells, after 4 h infection, epithelial cells were stained with a CellTracker Orange CMTMR (1:5000 dilution) (Thermo Fisher Scientific, Waltham, MA, USA) and *N. gonorrhoeae* was detected with Anti-*Neisseria gonorrhoeae* antibody (1:200) (Abcam, Cambridge, UK) as primary antibodies and Goat Anti-Mouse IgG H and L (1:500) conjugated with Alexa Fluor^®^ 488 (green-fluorescent dye) (Abcam) as secondary antibodies.

SCLM was performed using a Nikon Eclipse Ti (A1) microscope (Nikon, Tokyo, Japan) equipped with a ×60, 1.4 NA oil immersion phase-contrast lens and an argon laser with a maximum emission line at 565 nm for CellTracker Orange CMTMR and at 519 nm for Alexa Fluor^®^ 488. Horizontal optical thin sections were collected at 0.21 µm intervals from the outer surface of the biofilm or epithelial cells to the bottom of the glass plate. NIS-ELEMENTS interactive software was used for capture and three-dimensional (3D) reconstructions of images.

### 2.6. The Crystal Violet Assay

The crystal violet assay was performed as described previously [27,28,29,30] with slight modifications. Briefly, gonococcal cell suspensions (10^7^ cells/mL in GC broth) were added into a 96-well microtiter plate, and bacteria were grown without shaking for 24 h at 37 °C in 5% CO_2_. Then, the amounts of planktonic cells in each well was assessed by measuring OD600 followed by aspiration and removal of planktonic cells. The remaining attached bacteria were fixed with ethanol, air-dried, and stained with 0.8% crystal violet dye. Excess stain was rinsed off, samples were air-dried, and the crystal violet dye bound to the adherent cells was resolubilized with 96% ethanol. The amount of adhered bacterial cells stained by crystal violet was measured at 570 nm using an automated Sunrise microplate reader and Magellan software (Magellan 7.2, Tecan, Männedorf, Switzerland).

### 2.7. RNA Extraction

RNA was extracted from (i) human epithelial cells not infected by *N. gonorrhoeae*, (ii) human epithelial cells infected by *N. gonorrhoeae* as done in the adhesion assay (4 h infection), (iii) *N*. *gonorrhoeae* cultivated on a GC agar base for 24 h, and (iv) *N. gonorrhoeae* infecting human epithelial cells (4 h infection). Total RNA from the above samples was extracted using a High Pure RNA Isolation Kit (Roche, Basel, Switzerland). RNA from *N. gonorrhoeae* infecting human epithelial cells was separated from mixed samples containing human and bacterial RNA using a MICROBEnrich™ Kit (Thermo Fisher Scientific). Contaminating DNA was removed using DNA-free™, DNase Treatment and Removal (Thermo Fisher Scientific). RNA quality was measured using a 2100 Bioanalyzer (Agilent Technologies, Santa Clara, CA, USA), and RNA with a RIN (RNA integrity number) >8.0 was used.

### 2.8. Microarray Experiments

Detailed protocols and specifications for DNA microarray analysis and array analysis are described in Kwiatek et al. [27,28]. Briefly, for the microarray experiments, an Agilent custom-designated 60-mer oligonucleotide array (Agilent-034141) was used. RNA was extracted according to the protocol described above. Further manipulations with RNA (amplification and labeling whole transcripts with Cy3 and Cy5) were performed using the Low Input Quick Amp Labeling WT Kit (Agilent Technologies). The Cy3 dye was used to label the wild-type cRNA, and the Cy5 dye was used to label the cRNA of the *N. gonorrhoeae* mutants. To monitor the microarray workflow, as an internal control, the RNA Spike in Kit for Two Color v4.0 (Agilent Technologies) was used. Equal amounts of Cy5- and Cy3-labeled cRNAs were hybridized onto the microarray for 17 h at 65 °C using the Gene Expression Hybridization Kit (Agilent Technologies) followed by microarray washing with the Gene Expression Wash Buffer Kit (Agilent Technologies). Microarray slides were scanned with an Agilent G2565CA Microarray Scanner System (Agilent Technologies).

### 2.9. Array Analysis

Array analysis was performed using GeneSpring version 12.5 (Agilent Technologies). Normalized data were filtered on genes flagged as present or marginal, with the resulting gene list used for further gene expression analysis and clustering (Student’s *t*-test against zero, asymptotic *p* value computation, and Benjamini-Hochberg FDR multiple testing correction). Changes were expressed as the ratio of gene expression of the wild-type FA1090 strain over the expression of the same gene for the mutant. Genes with differential expression of ≥1.5-fold and a *p* value < 0.05 were selected. Supporting microarray data for all studied gonococcal mutants have been deposited in the NCBI’s Gene Expression Omnibus and are accessible through the GEO Series accession number GSE61789.

### 2.10. RT-qPCR

RT-qPCR experiments were performed according to the Minimum Information for Publication of Quantitative Real-Time PCR Experiments guidelines [31]. Bacterial cDNAs were obtained by reverse transcription of 10 µg of total RNA using the Maxima First Strand cDNA Synthesis Kit for RT-qPCR and human cDNAs using the RevertAid First Strand cDNA Synthesis Kit (Thermo Fisher Scientific). RT-qPCR using 1× HOT FIREPol^®^ EvaGreen^®^ qPCR Mix Plus (ROX) (Solis BioDyne, Tartu, Estonia) was performed on a LightCycler^®^96 system (Roche, Basel, Switzerland) (95 °C for 900 s, 40 cycles of 95 °C for 15 s, 60 °C for 30 s, and 72 °C for 20 s). Oligonucleotide primer pairs (HPLC purified) specific for each bacterial gene of interest were designed using PrimerQuest software (Primer 3 2.2.3, Cambridge, MA, USA) hosted by Integrated DNA Technologies and purchased from Sigma-Aldrich (Saint Louis, MO, USA). Oligonucleotide primer pairs specific for each human gene of interest were obtained from Bio-Rad (Hercules, CA, USA) or Thermo Fisher Scientific (Appendix A). Relative quantification of gene expression was performed using the comparative ΔΔ*C*t (threshold cycle) method. The relative amount of target cDNA was normalized using the *16S rRNA* gene as an internal reference standard for bacterial gene expression and both the Hypoxanthine phosphoribosyltransferase 1 (*HPRT 1*) gene and Beta-2-microglobulin gene for human gene expression.

### 2.11. Adhesion Assay

The adhesion assay was performed as described previously [27,28,32]. Briefly, confluent epithelial cells were infected by *N. gonorrhoeae* (m.o.i. 1:100, bacterial suspension (10^7^ cells/mL) made in DMEM) for 4 h at 37 °C in 5% CO_2_. Next, one set of the *N. gonorrhoeae*-infected epithelial cells was washed with PBS, lysed by 0.5% saponin, and lysates were plated on GC agar base with 1% hemoglobin to enumeration of CFU of cell-associated gonococci. At the same time, another set of the *N*. *gonorrhoeae*-infected epithelial cells wasn’t rinsed with PBS, but only culture medium from the above cells was removed, following this, epithelial cells were lysed by 0.5% saponin and plated on GC agar base with 1% hemoglobin to enumeration of CFU. The sum of CFU from the medium supernatant and from the cell lysates constitutes total CFU. The adhesion index was calculated by dividing the number of cell-associated CFU by the number of total CFU.

### 2.12. Invasion Assay

An invasion assay was performed as described previously with slight modifications [27,28,32]. Briefly, confluent epithelial cells were infected by *N. gonorrhoeae* (m.o.i. 1:100, bacterial suspension (10^7^ cells/mL) made in DMEM) for 4 h at 37 °C in 5% CO_2_, as for adhesion assays. Then, a set of the *N*. *gonorrhoeae*-infected epithelial cells was incubated with 0.2 mg/mL gentamicin for 1 h at 37 °C to kill extracellular gonococci and further lysed by 1% saponin to release intracellular gonococci. The lysates were plated on a GC agar base with 1% hemoglobin to enable enumeration of intracellular (gentamicin-resistant (Gm^R^)) *N. gonorrhoeae*. At the same time, another set of the *N*. *gonorrhoeae*-infected epithelial cells was processed for calculation of *N. gonorrhoeae* total CFU, as was described for the adhesion assay. The invasion index was calculated by dividing the number of Gm^R^ CFU by total CFU.

### 2.13. Computer Analysis, In Silico Analysis of Metabolic Pathways, and Determination of Clusters of Orthologous Groups (COG) Category

DNA and protein sequences were compared with GenBank on the BLAST server hosted by the National Center for Biotechnology Information (www.ncbi.nlm.nih.gov/blast), KEGG (Kyoto Encyclopedia of Genes and Genomes), and UniProt databases. Clusters of Orthologous Groups (COGs) were assigned with the NCBI Conserved Domains server (http://www.ncbi.nlm.nih.gov/Structure/cdd/wrpsb.cgi) [33,34].

Comstat2 software (http://www.comstat.dk, Lyngby, Denmark) was used for an analysis of image stacks of biofilms recorded by SCLM [35], which allowed for determination of biofilm biomass, roughness coefficient, and maximum thickness. According to this software, biofilm biomass is defined as volume by surface area of the image (μm^3^/μm^2^) and represents the volume of the biofilm in the image (volume/area or μm^3^/μm^2^). Roughness coefficient (R*_a_) is a measure for the variability in the height of the biofilm. In Comstat2, the number of thickness measurements equals the number of pixels in the substratum, as the thickness is determined for each pixel in the observed area.

### 2.14. Chemicals and Kits

Chemicals were purchased from Sigma-Aldrich unless otherwise noted. All methods performed with described kits were performed according to the manufacturer’s recommendations.

## 3. Results and Discussion

### 3.1. The Genes Encoding Proteins that can Influence Adhesion and Biofilm Formation Are Differentially Expressed in N. gonorrhoeae mutL::km and N. gonorrhoeae mutS::km Mutants Compared to the Wild-Type Strain

To identify processes, other than DNA repair, in which gonococcal MutL and MutS proteins may be involved, an analysis of global transcriptomes of gonococcal *mutL* and *mutS* mutants and the parental wild-type *N. gonorrhoeae* FA1090 cultivated on abiotic surfaces was performed using DNA expression microarrays (followed by RT-qPCR in selected cases). A total of 79 genes in the *N. gonorrhoeae mutL::km* mutant and 42 genes in the *N. gonorrhoeae mutS::km* mutant had a 1.5-fold cutoff threshold for differential expression compared to the parental wild-type strain. In the *N. gonorrhoeae mutL::km* and *N. gonorrhoeae mutS::km* mutants, 46 and 31 genes were upregulated respectively, and 33 and 11 were downregulated (Figure 1 and Appendix A).

Overall, upregulated genes in the *N. gonorrhoeae mutS::km* mutant encode proteins belonging to protein Cluster of Orthologous Groups (COG) categories including: cellular processes and signaling, metabolism, and poorly characterized or proteins without known domains, whereas proteins encoded by downregulated genes belonged only to cellular processes and signaling, metabolism, and poorly characterized. In turn, in the *N. gonorrhoeae mutL::km* mutant, both upregulated and downregulated genes encode proteins belonging to COG categories such as cellular processes and signaling, metabolism, information storage and processing, and poorly characterized or proteins without known domains (Figure 1 and Appendix A).

Based on the results of global transcriptomic analysis, we focused on selected genes encoding proteins that can influence adhesion and biofilm formation (Appendix A and Appendix A). The DNA microarray analysis indicated that in both the *N. gonorrhoeae mutL::km* and *N. gonorrhoeae mutS::km* mutants, there was an increase in the expression of *ngo1393* and *ngo1585* genes. These genes form a part of the *mafA-mafB-mafI* gene clusters located in so-called maf genomic islands (MGIs), which constitute approximately 2% of the genome of pathogenic *Neisseria* species [36]. The genome of *N. gonorrhoeae* FA1090 contains five genomic islands of this type (MGI-1–MGI-5). The *ngo1393* gene is associated with *ngo1392* gene and the *ngo1585* gene with the *ngo1584* gene. The genes *ngo1393*–*ngo1392* and *ngo1584*–*ngo1585* are located within MGI-5_FA1090_ and MGI-1_FA1090_, respectively. The two regions share a high level of nucleotide sequence similarity (Appendix A). In particular, the nucleotide sequences of *ngo1393* and *ngo1584* genes are 100% identical. Whereas, the *ngo1392* and *ngo1585* genes exhibit 99% similarity of the 1487-nt-long 5′-terminal part of their 1818-nt-long coding sequences. The gene expression analysis shows that the expression of the *ngo1393* gene was increased 2.70-fold and 8.14-fold in *N. gonorrhoeae mutL::km* and *N. gonorrhoeae mutS::km* respectively, compared to the parental strain (*p* = 1.34 × 10^−4^, 2.06 × 10^−5^, respectively). Whereas, the *ngo1585* gene was 2.70-fold and 4.90-fold overexpressed in *N. gonorrhoeae mutL::km* and *N. gonorrhoeae mutS::km* respectively, compared to the parental strain (*p* = 2.86 × 10^−4^, 6.37 × 10^−6^, respectively). The *ngo1392* and *ngo1585* genes encode MafB proteins whose homologue was demonstrated to function as a secreted polymorphic toxin with EndoU ribonuclease activity in *N. meningitidis* NEM8013 [36]. According to Jamet and Nassif, Jamet et al., and Arenas et al., meningococcal MafB toxins could be engaged in competition between different microorganisms occupying the same ecological niche [36,37,38]. Thus, our results demonstrating the increased expression of the *ngo1585* gene in gonococcal *mutL* and *mutS* mutants could imply that the gonococcal mutator phenotype ascribed to deficiency in *mutL* or *mutS* genes may be favored in interbacterial competition between microorganisms sharing the same environment. In turn, the products of the *ngo1393* and *ngo1584* genes are predicted to encode MafA proteins, which have been shown to function as glycolipid-binding adhesins [39] and have been found in the outer membrane and outer membrane vesicles (OMVs) [40]. Consequently, it has been suggested that the delivery of MafB toxins to bacterial or eukaryotic cells may be an OMVs-dependent process [36].

In addition to the *ngo1393*-*ngo1392* and *ngo1584*-*ngo1585* genes, in *N. gonorrhoeae mutL* and *mutS* mutants, several other genes encoding proteins that can influence adhesion, intraspecies communication, or biofilm formation were differentially expressed. In *N. gonorrhoeae mutL::km*, these genes are *ngo0206*, *ngo1535*, and *ngo1822* (*secY*), and in *N. gonorrhoeae mutS::km*, they are *ngo0626* and *ngo0834* (Appendix A and Appendix A).

The *ngo0206* gene, which expression was increased by 2.17-fold (*p* = 2.95 × 10^−5^) encodes a polyamine ABC transporter substrate-binding protein that according to the KEGG database, belongs to the quorum sensing system. So far, there are no data indicating that *N. gonorrhoeae* has a quorum sensing system. The only data about this system coded by *Neisseria* spp. concern LuxS (AI-2), encoded by *N. meningitidis*, which is thought to be a metabolic by-product with a yet undefined role in quorum sensing [41]. However, we cannot rule out that the product of the *ngo0206* gene in *N. gonorrhoeae* is engaged in intraspecies communication that allows bacteria to communicate about cell density. The *ngo1535* (*murD*) gene, whose expression was increased 2.45-fold (*p* = 3.97 × 10^−4^) in the gonococcal *mutL* mutant compared to the parental strain (Appendix A), encodes MurD, a UDP-N-acetylmuramoyl-l-alanyl-d-glutamate synthetase (EC 6.3.2.9) involved in peptidoglycan biosynthesis during cell wall formation. Falsetta et al. demonstrated that this gene was differentially expressed in biofilm cells compared to planktonic cells [42]. In turn, the *ngo1822* (*secY*) gene encodes the preprotein translocase SecY belonging to the COG0201:SecY category, which is involved in intracellular trafficking, secretion, and vesicular transport. Deficiency in biofilm formation by *Streptococcus pneumoniae* TIGR4 with a disrupted *secA* gene encoding preprotein translocase [43] could indirectly suggest that the 1.80-fold (*p* = 2.22 × 10^−4^) increased expression of the *ngo1822* (*secY*) gene in the *mutL* gonococcal mutant may influence biofilm formation by *N. gonorrhoeae mutL::km*.

The *ngo0834* gene, whose expression is increased 4.72-fold (*p* = 1.86 × 10^−6^) in *N. gonorrhoeae mutS::km* compared to expression in the parental strain (Appendix A), encodes a membrane protein that is a component of curli, which are biologically important cell surface fibers associated with biofilm formation and composition, adhesion to host cells and colonization, as was demonstrated for Bacteroidetes and Proteobacteria, e.g., for *E. coli*, *Salmonella* sp., and *Caulobacter crescentus* [44,45,46,47,48,49,50,51,52]. The *ngo0626* gene, overexpressed 1.89-fold in *N. gonorrhoeae mutS::km* (*p* = 3.36 × 10^−6^) encodes lytic transglycosylase (LT), and recently, it was demonstrated that in addition to the role of LTs in peptidoglycan turnover, these enzymes facilitate pathogenesis in *N. gonorrhoeae* [53,54]. Moreover, in *Salmonella enterica* serovar Typhimurium, in addition to engagement in cell wall turnover, LTs participate in signaling pathways that link cell wall turnover to biofilm formation [55].

Thus, the differential expression of the above genes can be relevant for adhesion and biofilm formation by *mutL* and *mutS* mutants, since in addition to pilus type IV and Opa proteins, *N. gonorrhoeae* also has other surface adhesins [39]. It is also worth mentioning that identification of COGs was possible for differentially expressed genes of only 66 of 79 proteins and 31 of 42 proteins for the *N. gonorrhoeae mutL::km* and *N. gonorrhoeae mutS::km* mutants, respectively. The remaining proteins encoded by genes with changed expression did not exhibit any known conserved domains; therefore, we cannot rule out that among these proteins, there are also those that participate in adhesion and biofilm formation.

In *Neisseria* spp., many genes undergo phase variation, and meningococcal MMR proteins are related to this phenomenon [15,56]. Therefore, to study whether the differential expression demonstrated for genes in *N. gonorrhoeae mutL* and *mutS* mutants might result from phase variation, we examined upstream and within sequences of differentially expressed genes with respect to the occurrence of homo- and hetero-polymeric tracts, which could be phase variable. Our in silico analysis ruled out the presence of such repetitions upstream and within sequences of most differentially expressed genes, with the exception of *ngo0207* and *ngo0721* in the *N. gonorrhoeae mutL* mutant. A more detailed analysis of the upstream sequences of differentially expressed genes, specifically gonococcal *maf* genes, identified the presence of a common nucleotide motif resembling the NadR minimal binding sequence (5′-AATCCGTTCAACATCAAACAA-3′). Further, we discovered similar sequences upstream of other differentially expressed genes in gonococcal *mutL* and *mutS* mutants (Appendix A). Thus, the observed differential expression of MGI-associated genes may be dependent on the activity of the NadR regulon in the analyzed mutants. It is worth noting that gonococcal MafA proteins NGO1393 and NGO1584 exhibit 97% identity to the product of the *mafA1* gene (AAF62309.1) of *N. meningitidis* MC58, whose expression was shown to be regulated by the action of the transcriptional regulator NadR [57].

### 3.2. Inactivation of MutL or MutS Genes Changes the Formation and Structure of the Biofilm by N. gonorrhoeae on Abiotic Surfaces

Considering the differential expression of certain genes encoding proteins that can influence adhesion and biofilm formation, the biofilm-forming ability by *N. gonorrhoeae mutL::km* and *N. gonorrhoeae mutS::km* mutants on abiotic surface was determined and compared to those formed by the wild-type strain using the crystal violet assay, SCLM, and FE SEM. Both the wild-type strain and mutants adhered to the abiotic surface and generated visible biofilms.

We have observed that the *mutL* and *mutS* mutants produced more biofilm-associated biomass, 1.26-fold (*p* = 4.42 × 10^−6^) and 1.52-fold (*p* = 1.12 × 10^−8^) respectively, than did the wild-type parental strain, as was determined by the crystal violet assay. Simultaneously, the increase in the number of cells forming the biofilm was accompanied by a decrease in the number of cells in the planktonic stage, suggesting that *mutL* and *mutS* mutants have a greater capability to form biofilms than wild-type parental strains (Figure 2).

The comparison of the three-dimensional structure of live biofilm revealed that the *N. gonorrhoeae mutS::km* mutant forms more condensed biofilms than those observed for the wild-type strain, as demonstrated by SCLM and quantified using a Comstat2 software (Figure 3 and Appendix A). The average biomass of biofilm formed by *N. gonorrhoeae mutS::km* mutant was 3.19-fold higher than the biomass of the wild-type strain biofilm (15.18 ± 2.85 µm^3^/µm^2^ versus 4.75 ± 0.47 µm^3^/µm^2^; *p* = 2.32 × 10^−4^). *N. gonorrhoeae mutL::km* also formed a biofilm whose average biomass was higher than the biomass of the biofilm created by the wild-type strain. However, the increase was only 1.25-fold (average biomass 5.93 ± 0.99 µm^3^/µm^2^ versus 4.75 ± 0.47 µm^3^/µm^2^; *p* = 3.26 × 10^−2^) (Figure 3). The roughness coefficient (R*_a_), which is an indicator of biofilm heterogeneity reflecting diversity in thickness and biomass through a biofilm, was significantly different both for *N. gonorrhoeae mutS::km* biofilm and for *N. gonorrhoeae mutL::km* biofilm compared to that of the wild-type strain (R*_a_ 0.37 ± 0.078 and 0.92 ± 0.12 versus 0.75 ± 0.095, respectively; *p* = 2.81 × 10^−5^ and 2.42 × 10^−2^, respectively). Furthermore, biofilm-forming cells of *N. gonorrhoeae mutL::km* were arranged in cell clusters and formed characteristic roughness (Appendix A). It could indicate that the structure of biofilms formed by the *N. gonorrhoeae mutL::km* and *N. gonorrhoeae mutS::km* mutants differs from that of the wild-type strain biofilm. However, the maximum thickness of biofilms formed by *N. gonorrhoeae mutL::km*, and *N. gonorrhoeae mutS::km* and wild-type strain equaled 27.16 ± 4.27 µm (data not presented).

Accordingly, the higher biofilm-forming cell density and more condensed biofilms formed by *N. gonorrhoeae mutL::km* and *N. gonorrhoeae mutS::km* mutants compared to the wild-type strain were also visible when FE SEM were used (Appendix A).

It is also noteworthy that, although both mutants produced more biofilm-associated biomass, gonococcal *mutS* mutant formed biofilm with higher average biomass than the *mutL* mutant. Based on our results, we can suppose that it may be related to different gene expression in gonococcal *mutL* and *mutS* mutants. 79 genes in the *N. gonorrhoeae mutL::km* mutant and 42 genes in the *N. gonorrhoeae mutS::km* mutant were differentially expressed compared to the parental wild-type strain. An additional analysis revealed that *N. gonorrhoeae mutL::km* and *N. gonorrhoeae mutS::km* mutants share only five differentially expressed genes. Among them, there are *ngo1393* and *ngo1585* genes which encode proteins that may be engaged in interactions between gonococci or with their host. However, it should be noticed that the expression of these genes was significantly increased in the *mutS* mutant compared to the *mutL* mutant (Appendix A). Moreover, 16 and 12 genes, in the *mutL* and *mutS* mutants respectively, encode proteins that were not determined to any COG domain. Therefore, we cannot rule out that some of these genes encode proteins that may also participate in biofilm formation or these involved in the interactions with human host. Nevertheless, further examination is required in order to determine the precise functions of proteins encoded by these genes.

Introduction *in cis* of wild-type copies of *mutL* or *mutS* genes in the genome of appropriate *N. gonorrhoeae* mutants resulted in restoration of the level of adherence, biofilm formation, and phenotype of biofilms comparable to wild-type gonococcus strains.

### 3.3. N. gonorrhoeae mutS::km but not N. gonorrhoeae mutL::km Is Characterized by Increased Adhesion and Invasiveness to Human Epithelial Cells Compared to the Wild-Type Strain

As shown above, the *N. gonorrhoeae mutL::km* and *N. gonorrhoeae mutS::km* mutants have increased adhesion and biofilm formation ability on abiotic surfaces. Therefore, the adhesion and invasiveness of these mutants to human epithelial cells was studied by determining adhesion and invasion indexes. A comparison of the adhesion indexes of *N. gonorrhoeae mutS::km* and the wild-type strain showed that the mutant has 2.42-fold higher adhesion to human epithelial cells than wild-type strain (adhesion index 3.77 × 10^−1^ ± 3.59 × 10^−2^ versus 1.56 × 10^−1^ ± 2.85 × 10^−2^; *p* = 1.34 × 10^−13^) (Figure 4). Whereas, the adhesion index of *N. gonorrhoeae mutL::km* did not significantly differ compared to those of the wild-type strain (adhesion index 1.20 × 10^−1^ ± 2.23 × 10^−2^ versus 1.56 × 10^−1^ ± 2.85 × 10^−2^; *p* = 6.05 × 10^−1^) (Figure 4).

Microscopy visualization of epithelial cells infected by the *N. gonorrhoeae mutL::km*, *N. gonorrhoeae mutS::km*, and wild-type strains confirmed the presence of diplococci and small microcolonies of each strain on the surface of human epithelial cells (Figure 5 and Appendix A). Furthermore, analogously, as was determined by comparison of the adhesion indexes, an increased amount of *N. gonorrhoeae mutS::km* attached to human epithelial cells was detected compared to the wild-type strain, whereas there were no differences in the amount of bacteria attached to human cells between *N. gonorrhoeae mutL::km* and the wild-type strain.

The results presented in Figure 4 indicate that *N. gonorrhoeae mutS::km* is characterized not only by increased adhesion but also by increased invasiveness, as demonstrated by a 3.04-fold increase in the invasion index of the mutant compared to the wild-type strain (invasion index 1.65 × 10^−5^ ± 2.59 × 10^−6^ versus 5.43 × 10^−6^ ± 1.19 × 10^−6^; *p* = 6.62 × 10^−10^). This effect was not observed for *N. gonorrhoeae mutL::km*, whose invasiveness was not significantly different from that of the wild-type strain (invasion index 5.60 × 10^−6^ ± 1.25 × 10^−6^ versus 5.43 × 10^−6^ ± 1.19 × 10^−6^; *p* = 7.32 × 10^−1^) (Figure 4).

The adhesion and invasiveness of *N. gonorrhoeae* complementation mutants to epithelial cells were not significantly different from that of the wild-type strain. Adhesion indexes for these gonococci were as follows: 1.33 × 10^−1^ ± 2.63 × 10^−2^ and 1.69 × 10^−1^ ± 2.88 × 10^−2^ for *N. gonorrhoeae mutL::km* + *mutL_wt_* and *N. gonorrhoeae mutS::km* + *mutS_wt_* respectively, versus 1.56 × 10^−1^ ± 2.85 × 10^−2^ for wild-type strain (*p* = 5.80 × 10^−2^ and *p* = 2.68 × 10^−1^, respectively). Invasion indexes for these gonococci were as follows: 4.96 × 10^−6^ ± 1.56 × 10^−6^ and 5.86 × 10^−6^ ± 1.37 × 10^−6^ for *N. gonorrhoeae mutL::km* + *mutL_wt_* and *N. gonorrhoeae mutS::km* + *mutS_wt_* respectively, versus 5.43 × 10^−6^ ± 1.19 × 10^−6^ wild-type strain (*p* = 4.17 × 10^−1^, *p* = 4.22 × 10^−1^, respectively).

### 3.4. N. gonorrhoeae mutS::km Has an Altered Expression of Genes Encoding Proteins that can Influence Adhesion to Human Epithelial Cells

The *N. gonorrhoeae mutS::km* exhibits increased adhesion to human epithelial cells and differential expression of genes encoding proteins that can participate in adhesion and biofilm formation during growth on the abiotic surface. Accordingly, the expression of *ngo0626*, *ngo0834*, *ngo1393*, and *ngo1585* genes during the growth of *N. gonorrhoeae mutS::km* on human epithelial cells was evaluated. In contrast to the upregulation of these genes in the mutant grown on the abiotic surface, the expression of *ngo0626*, *ngo0834*, and *ngo1585* genes in the mutant cultivated on epithelial cells was decreased in comparison to the wild-type strain also cultivated on epithelial cells (1.42-, 1.3-, and 2.7-fold, respectively; *p* = 2.47 × 10^−2^, 4.63 × 10^−3^ and 4.75 × 10^−3^, respectively). Whereas, the expression of the *ngo1393* gene in the mutant was similar to that in the wild-type strain (*p* = 3.36 × 10^−1^). Interestingly, during infection of epithelial cells with *N. gonorrhoeae mutS::km*, the expression of the *ngo1535* gene was increased 3-fold compared to that in the wild-type strain infecting epithelial cells (*p* = 2.38 × 10^−4^) (Figure 6).

The upregulation of the *ngo1535* gene in the mutant compared to the parental strain can explain the increased adhesion of *N. gonorrhoeae mutS::km* to epithelial cells, because this gene is downregulated in the planktonic growth of the wild-type strain [42]. Differences in the expression of the above genes in the mutant grown on abiotic and biotic surfaces could imply that there is a different molecular basis for increased adhesion and biofilm formation by the *N. gonorrhoeae mutS::km* mutant on these surfaces. The upregulation of the *ngo1535* gene appears to be particularly important for the growth of the *N. gonorrhoeae mutS::km* mutant on epithelial cells since the expression of this gene was decreased 1.4-fold when the mutant was grown on an abiotic surface but increased 3-fold when gonococcus was cultivated on epithelial cells.

### 3.5. During Infection of Human Epithelial Cells with N. gonorrhoeae mutS::km, the Expression of Some Host Cell-Surface Receptors is Increased

The network of interactions between *N. gonorrhoeae* and the host involves not only bacterial proteins but also host cell-surface receptors. Accordingly, the expression of genes encoding host cell-surface receptors for *N. gonorrhoeae* [58,59], such as the complement regulator CD46, heparin sulfate proteoglycan 2 (HSPG 2), carcinoembryonic antigen-related cell adhesion molecule 1 (CEACAM 1) receptor, fibronectin, and vitronectin, in epithelial cells infected with the *N. gonorrhoeae mutS::km* mutant was estimated and compared with the expression in epithelial cells infected with wild-type *N. gonorrhoeae*.

The expression of genes encoding host-cell surface receptors such as CD46, HSPG 2, and CEACAM 1 was increased 2.12-, 1.67- and 2.37-fold (*p* = 4.19 × 10^−5^, 1.39 × 10^−5^ and 2.41 × 10^−5^, respectively) in cells infected with the mutant compared to those infected with the wild-type strain (Figure 7) after 4 h of infection. In contrast, the expression of genes encoding fibronectin and vitronectin was at the same level in epithelial cells infected with both the *N. gonorrhoeae mutS::km* mutant and the parental strain (*p* = 2.96 × 10^−1^ and 3.16 × 10^−1^, respectively). Since CD46, HSPG 2, and CEACAM 1 are utilized by gonococci to attach and adhere to host cells, the increased expression of their genes could support the increased adhesion of the mutant to host cells.

## 4. Conclusions

The mutator phenotype ascribed to deficiency of the *mutS* gene results in not only increased mutation rates but also increased biofilm formation, adhesion, and invasiveness. Thus, although the mutator phenotype results in increased mutation frequency in singular organisms, mutator strains in the population can be beneficial because they can be a source of features linked to microbial fitness to host or environmental conditions [60]. We also hypothesize that deficiency of the *mutS* gene may also impact on virulence of this strain; however, understanding the molecular basis of this process requires further analysis, including determination of the functions of numerous unknown bacterial genes whose expression is affected by *mutS* and *mutL* genes deletion, in order to clarify the role of their products in gonococcal physiology.

## Figures and Tables

**Figure 1 microorganisms-07-00647-f001:**
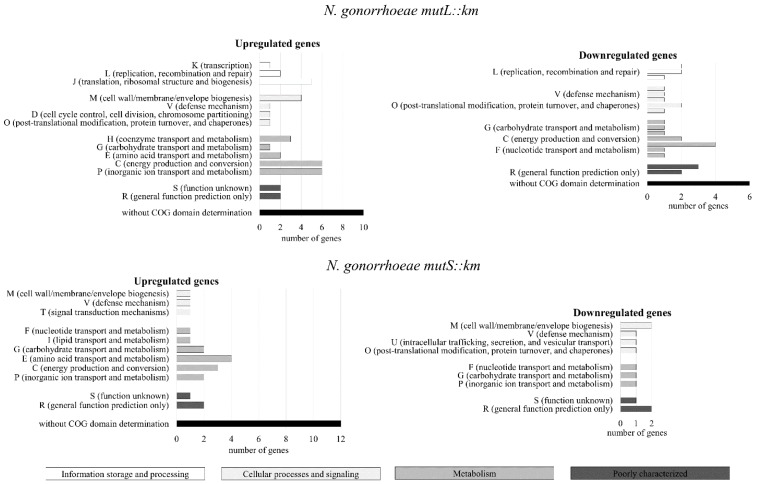
Differential gene expression in *Neisseria gonorrhoeae* (*N. gonorrhoeae*) *mutL* and *mutS* mutants determined by DNA microarray method. Classification of the Cluster of Orthologous Groups (COGs) by functional categories of proteins encoded by differentially expressed genes in *N. gonorrhoeae mutL* and *mutS* mutants cultivated on abiotic surface. The graphs show functional COGs categories. The color of the functional category corresponds to the main category indicated in rectangles.

**Figure 2 microorganisms-07-00647-f002:**
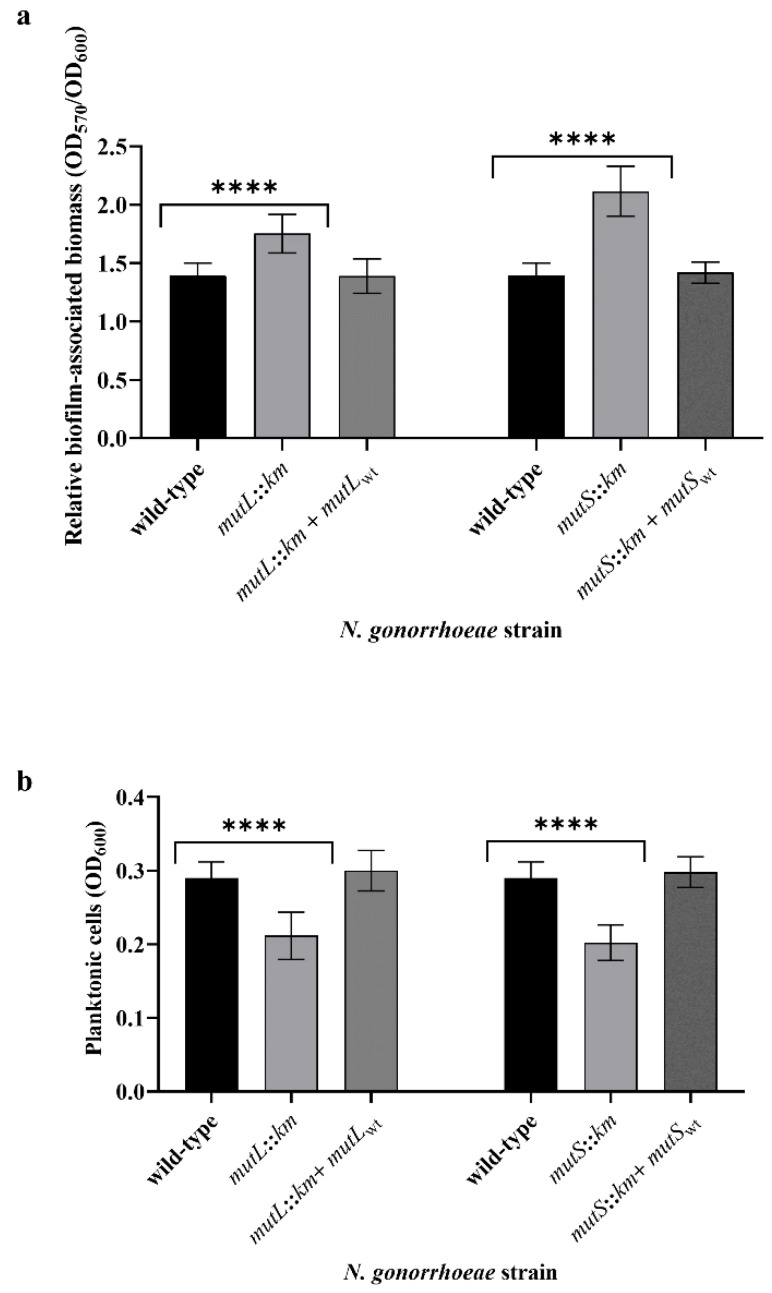
Effects of the inactivation of *mutL* and *mutS* genes on biofilm formation by *N. gonorrhoeae* on abiotic surfaces determined by the crystal violet assay. (**a**) The relative biofilm-associated biomass after 24 h. (**b**) The amount of gonococcal planktonic cells after 24 h. The asterisks indicate the statistically significant differences between mutants with a disrupted *mutL* or *mutS* gene (*N. gonorrhoeae mutL::km* and *N. gonorrhoeae mutS::km*, respectively), complementation strains (*N. gonorrhoeae mutL::km* + *mutL_wt_* and *N. gonorrhoeae mutS::km* + *mutS_wt_*) and the wild-type strain, as were calculated using two-way analysis of variance (ANOVA) followed by Bonferroni post-tests. Data represent mean values (± standard deviation (SD)) from three independent experiments performed in triplicate. **** asterisks indicate *p* values < 0.0001.

**Figure 3 microorganisms-07-00647-f003:**
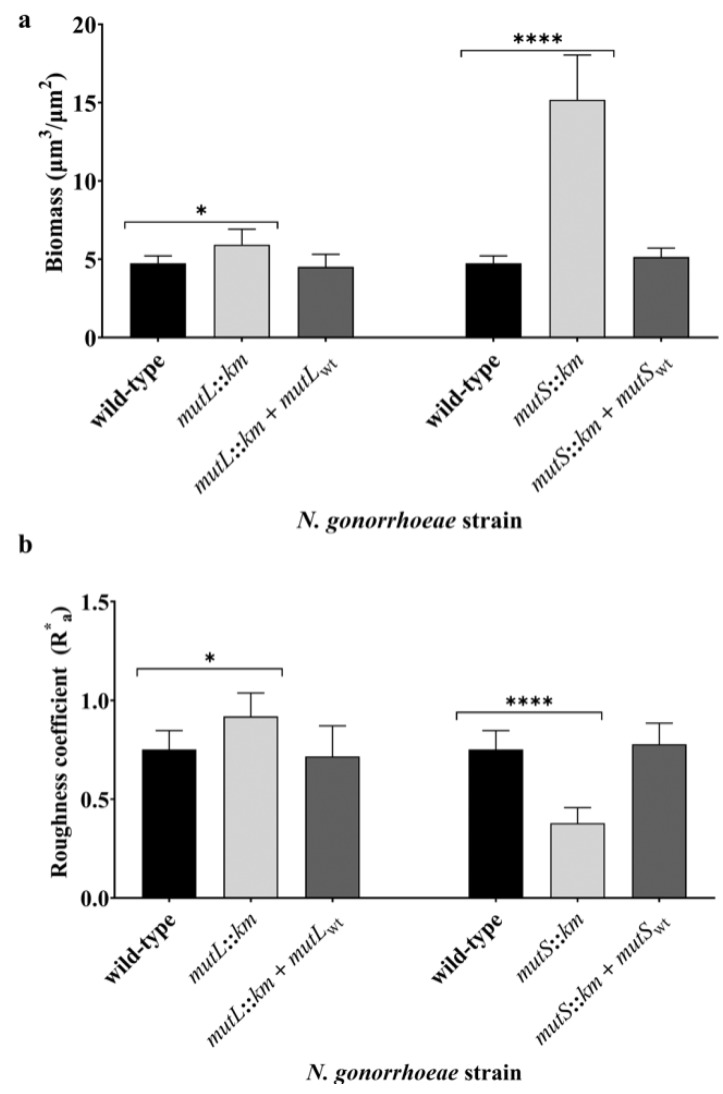
The structure and density of live biofilms formed by gonococci with a disrupted *mutL* or *mutS* gene (*N. gonorrhoeae mutL::km* and *N. gonorrhoeae mutS::km*, respectively), complementation strains (*N. gonorrhoeae mutL::km* + *mutL_wt_* and *N. gonorrhoeae mutS::km* + *mutS_wt_*), and the wild-type strain on abiotic surfaces assayed by scanning confocal laser microscopy (SCLM). Analysis of biofilms properties: (**a**) biomass of biofilm (µm^3^/µm^2^) and (**b**) roughness coefficient (R*_a_) determined by Comstat2 software. The statistically significant differences were calculated using two-way ANOVA followed by Bonferroni post-tests. Data represent mean values (± SD) from three independent experiments performed in triplicate. * and **** asterisks indicate *p* value < 0.05, and < 0.0001, respectively.

**Figure 4 microorganisms-07-00647-f004:**
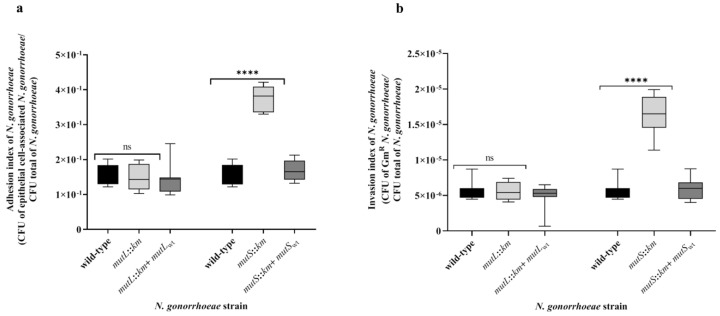
Adhesion and invasiveness of *N. gonorrhoeae mutL::km* and *N. gonorrhoeae mutS::km* to human epithelial cells. (**a**) The adhesion of *N. gonorrhoeae mutL::km* and *N. gonorrhoeae mutS::km* in comparison to complementation and the wild-type strains. (**b**) The invasiveness of *N. gonorrhoeae mutL::km* and *N. gonorrhoeae mutS::km* in comparison to complementation and the wild-type strains. The asterisks indicate statistically significant differences between mutants with a disrupted *mutL* or *mutS* gene (*N. gonorrhoeae mutL::km* and *N. gonorrhoeae mutS::km*, respectively), complementation strains (*N. gonorrhoeae mutL::km* + *mutL_wt_* and *N. gonorrhoeae mutS::km* + *mutS_wt_*), and the wild-type strain, as calculated using two-way ANOVA followed by Bonferroni post-tests. Data represent mean values (± SD) from three independent experiments performed in triplicate. **** asterisks indicate *p* values < 0.0001, ns—not statistically significant.

**Figure 5 microorganisms-07-00647-f005:**
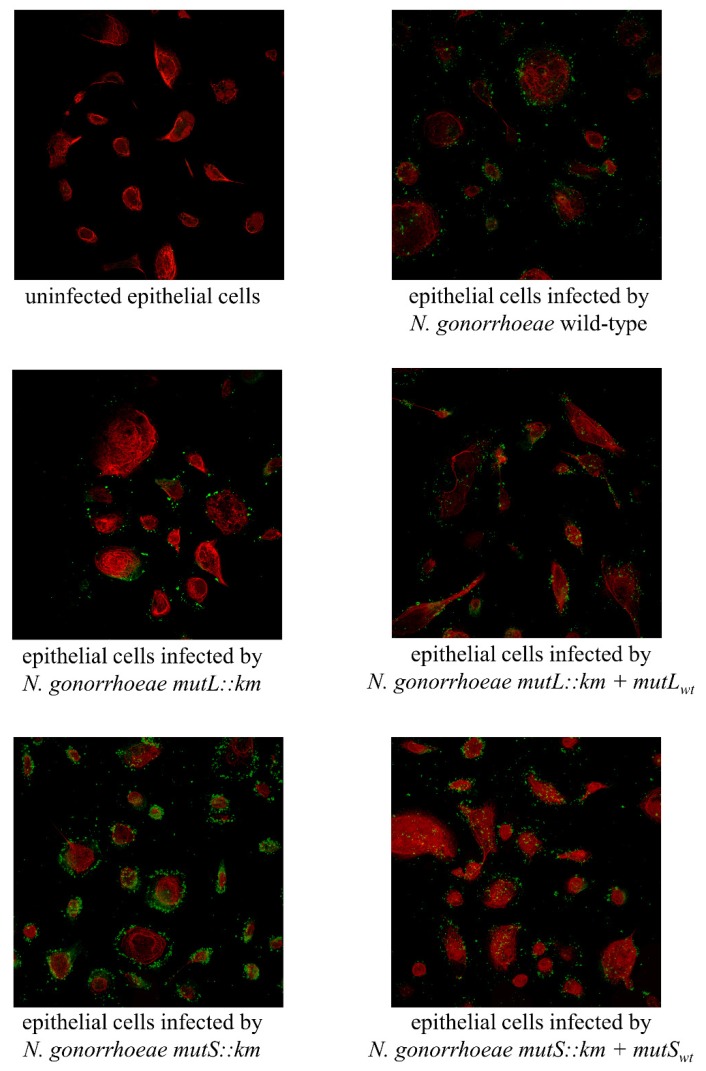
Microscopy visualization of epithelial cells infected by *N. gonorrhoeae mutL::km* and *N. gonorrhoeae mutS::km* assayed by SCLM (magnification: 40×). *N. gonorrhoeae* was added to the epithelial cells and cultivated for 4 h, followed by which, the human cells were stained with CellTracker Orange CMTMR, and *N. gonorrhoeae* were detected by Anti-*Neisseria gonorrhoeae* antibody as the primary antibody and Goat Anti-Rabbit IgG H and L conjugated with Alexa Fluor^®^ 488 (green-fluorescent dye) as the secondary antibody. All experiments were performed in triplicate, and representative images are shown.

**Figure 6 microorganisms-07-00647-f006:**
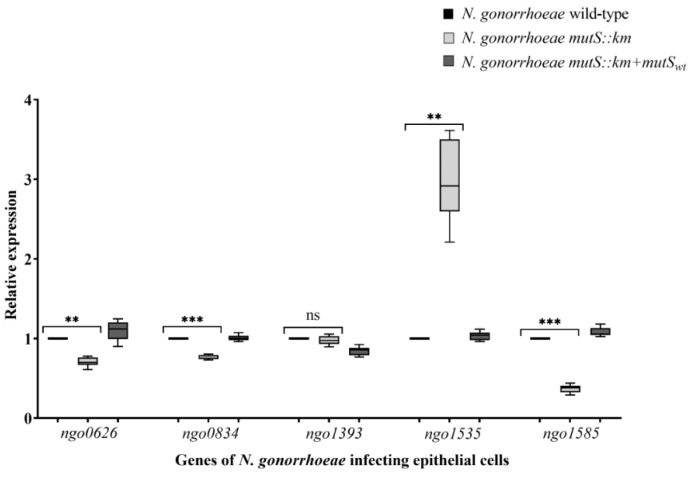
The expression of gonococcal *ngo0626*, *ngo0834*, *ngo1393*, *ngo1535*, and *ngo1585* genes during growth of *N. gonorrhoeae mutS::km* on human epithelial cells. The expression of the genes was evaluated by RT-qPCR using RNA extracted from bacterial cells infecting human epithelial cells. The relative quantitation data analysis was performed using the comparative quantification method ΔΔCt with *16S rRNA* as the endogenous reference. The statistically significant differences between *N. gonorrhoeae mutS::km*, the complementation strain (*N. gonorrhoeae mutS::km* + *mutS_wt_*), and the wild-type strain were calculated using two-way ANOVA followed by Bonferroni post-tests. Data on gene expression in *N. gonorrhoeae mutS::km* and *N. gonorrhoeae mutS::km* + *mutS_wt_* were compared to data on gene expression in the wild-type strain, for which the value of 1 was assumed. Data represent mean values (± SD) from three independent experiments performed in triplicate. **, *** asterisks indicate *p* values < 0.01 and < 0.0001 respectively, ns—not statistically significant.

**Figure 7 microorganisms-07-00647-f007:**
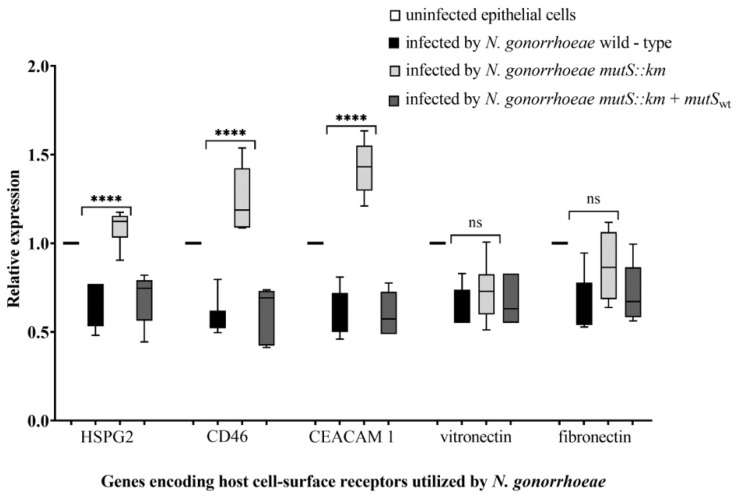
The expression of human genes encoding complement regulator CD46, heparin sulfate proteoglycan 2 (HSPG 2), carcinoembryonic antigen-related cell adhesion molecule 1 (CEACAM 1) receptor, fibronectin, and vitronectin in epithelial cells infected with *N. gonorrhoeae mutS::km* in comparison to those infected with wild-type *N. gonorrhoeae*. The expression of the genes was evaluated by RT-qPCR using RNA extracted from human epithelial cells infected 4 h with *N. gonorrhoeae*. The relative quantitation data analysis was performed using the comparative quantification method ΔΔ*C*t with *HPRT 1* (Hypoxanthine-guanine phosphoribosyltransferase 1) and Beta-2-microglobulin as endogenous references. The statistically significant differences in gene expression caused by *N. gonorrhoeae mutS::km* infection in comparison to infection caused by the complementation strain (*N. gonorrhoeae mutS::km* + *mutS_wt_*) and wild-type strain were calculated using two-way ANOVA followed by Bonferroni post-tests. Data were normalized to data from uninfected epithelial cells, for which the value of 1 was assumed. Data represent mean values (± SD) from three independent experiments performed in triplicate. **** asterisks indicate *p* values < 0.0001, ns—not statistically significant.

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
