# Peer review of "Inactivation of Genes Encoding MutL and MutS Proteins Influences Adhesion and Biofilm Formation by Neisseria gonorrhoeae"

_microorganisms, 2019, doi:10.3390/microorganisms7120647_

Round 1
Reviewer 1 Report
The paper by Placzkiewica et al described the effects of gene expression in Neisseria gonorrhoeae (Ng) FA1090 after the genes were mutated. The authors performed microarray experiments to study gene expression in mutL and mutS mutant strains and determined that 79 and 43 genes, respectively had different regulation from the wild type strain (grown without antibiotics). The authors then focussed on genes implicated in adhesion and biofilm formation and found an increase in expression of genes implicated in MafA and Maf B expression as well as 4 other genes. The authors ascertained from a bioinformatic analysis that the differential expression of certain of these genes could be due to regulation by NadR. This research also established that both mutant strains produced more biofilm-associated biomass than the wild type and that the MutS::km mutant produced a higher average biomass. These studies were supported by SEM data. Altered phenotypes of mutated strains could be reversed by complementing the mutant strains with a wild type gene. The authors also showed that the mutS::km but not the mutL::km mutant increased invasiveness and adhesion of Ng on human epithelial cells (HEC-1-B). These results were also verified by microscopic observation. Genes upregulated after growth on abiotic medium in the mutS::km strain were not upregulated in mutants grown on the HECs except for ngo1535. The authors also determined that infected HEC cells had higher expression of host cell receptors for NG. These are new findings, somewhat preliminary, which promise a number of follow-up experiments to clarify the conclusions postulated. The paper is nicely written, the data presentations are clear and the methods appropriate.
Some minor comments:
Line 60-61. Meaning not clear.
Line84. Why was Ng grown on hemoglobin containing medium prior to harversting for research. The rationale is not clear?
Line 243-247. Delete. Repetitive as such statements are also at eth end of the Introduction.
Line251 and beyond. The data in Supplementary Figs S1 and Table S2 should be indicated much earlier in the paragraph. In fact, I would include Fig S1 as a non-supplementary figure.
Throughout the text, N. gonorrhoeae must be written in italics. See Results and Discussion as well as lines 89-91).
Figure 1 was extremely difficult to read. The photos in Figure 2 could be larger/clearer and perhaps placed in supplementary data. Figure 3 could be supplementary data as well.
Figure 5 – it was difficult to see the Ng cells in the print out. Perhaps these pictures could be enlarged?
Can the authors postulate as to why the mutant strains differ with respect to average biomass and roughness coefficient?
Can the authors explain why the mutL mutant was not implicated in adhesion and invasiveness of HECs?
The authors might discuss some of the limitations of their work as well as future research that might be undertaken to further the impact of these findings.
Author Response
Thank you for all comments. Undoubtedly, these comments have improved the text clarity.
Line 60-61. Meaning not clear.
Our response: To clarify this sentence, we have corrected it as follow: ”Further, the meningococcal mutS mutant can be regarded as more pathogenic, because it has higher ability to escape from the bactericidal activity of monoclonal antibodies”. Please refer lines 60-61.
Line 84. Why was Ng grown on hemoglobin containing medium prior to harvesting for research. The rationale is not clear?
Our response: Neisseria gonorrhoeae is fastidious to growth in laboratory conditions and need rich medium. Medium supplemented with hemoglobin is recommended as solid medium for N. gonorrhoeae cultivation by ATCC (ATCC Medium: 814 GC Agar/Broth Medium). Nevertheless, media without hemoglobin (gonococcal base medium (GCB)) or a rich chemically defined medium, such as Graver-Wade medium are used (Dillard, 2011). We used GCB medium without hemoglobin to ensure about the pilliated state and Opa+ phenotype of gonococci, but for further manipulations we used to maintain Neisseria on GCB supplemented with both, hemoglobin and Kellogg.
P. Dillard, Genetic Manipulation of Neisseria gonorrhoeae. Current protocols in microbiology Chapter 4, Unit4A.2-Unit4A.2 (2011).
Line 243-247. Delete. Repetitive as such statements are also at eth end of the Introduction.
Our response: The referee is right. We have deleted these statements and for this reason we have also accordingly modified the first sentence of the paragraph. Line 259.
Line 251 and beyond. The data in Supplementary Figs S1 and Table S2 should be indicated much earlier in the paragraph. In fact, I would include Fig S1 as a non-supplementary figure.
Our response: We have included Fig. S1 as a non-supplementary figure. Currently, this figure is Figure 1. Please note that after correction, the numeration and legends of figures have been changed (both in non-supplementary and supplementary). According Your suggestion, current Figure 1 and Table S2 are indicated earlier in the paragraph. Please refer to line 266.
Throughout the text, N. gonorrhoeae must be written in italics. See Results and Discussion as well as lines 89-91.
Our response: The manuscript was edited and corrected.
Figure 1 was extremely difficult to read. The photos in Figure 2 could be larger/clearer and perhaps placed in supplementary data. Figure 3 could be supplementary data as well.
Our response: We have changed figures as follows:
Figure 1 – We have changed type of graphs. As Fig. S1 was included as a non-supplementary Figure 1, previous Figure 1 is now Figure 2.
Figure 2 – We have enlarged graphs and images and we have changed the type of graphs. Due to, as the referee recommended, we have divided the figure into two parts and redrawn them. Graphs are presented in Figure 3, and images of biofilms visualized by SCLM are included in Supplementary Fig. S3.
Figure 3 - We have included Fig. 3 as a supplementary figure, now this figure is Supplementary Fig. S4.
Figure 5 – it was difficult to see the Ng cells in the print out. Perhaps these pictures could be enlarged?
Our response: We have enlarged images.
Can the authors postulate as to why the mutant strains differ with respect to average biomass and roughness coefficient?
Our response: Based on our results, we can suppose that, it may be related to different gene expression in gonococcal mutL and mutS mutants. 79 genes in the N. gonorrhoeae mutL::km mutant and 43 genes in the N. gonorrhoeae mutS::km mutant were differentially expressed compared to the parental wild-type strain. An additional analysis revealed that N. gonorrhoeae mutL::km and N. gonorrhoeae mutS::km mutants share only 5 differentially expressed genes. Among them, there are ngo1393 and ngo1585 genes which encode proteins that may be engaged in interactions between gonococci or with their host. However, it should be noticed, that the expression of these genes was significantly increased in mutS mutant compared to mutL mutant (Table S2). Moreover, 16 and 12 genes, in mutL and mutS mutants, respectively, encode proteins that were not determined to any COG domain. Therefore we cannot rule out that some of these genes encode proteins that may also participate in biofilm formation or these involved in the interactions with human host. Nevertheless further examination is required in order to determine precise functions of proteins encoded by these genes.
We included such paragraph in text, lines 468-481.
Can the authors explain why the mutL mutant was not implicated in adhesion and invasiveness of HECs?
Our response: In our opinion, the difference between mutL and mutS mutants in adhesion and invasiveness of HECs can results from different gene expression in these mutants, as we have mentioned above.
The authors might discuss some of the limitations of their work as well as future research that might be undertaken to further the impact of these findings.
Our response: In our opinion, the limitations of the work as well as future research that might be undertaken to further is connected with more precisely understanding of molecular basis of described processes. Such analysis could include determination of the functions of numerous unknown bacterial genes that expression is affected by mutS and mutL genes disruption, in order to clarify the role of their products in gonococcal physiology.
We have completed the text. Please refer lines 700-704.
The manuscript was edited for grammar/English. We have obtained the certificate form AJE (https://www.aje.com) (verification code: 3716-99FB-D00E-72A8-A1CD).
Reviewer 2 Report
In this manuscript (Microorganisms 636408), authors studied the different gene expression in Neisseria gonorrhoeae mutL and mutS mutants using DNA microarrays.
Comments:
Authors should specify what type of bacterial strain is Neisseria gonorrhoeae FA1090Delete these paragraphs, please, put only the reference in parentheses: Page 2, line 77: in Dillard J.P. (2011) Page 3, line 103: in Adamczyk-Poplawska M. et al (2018) Page 3, line 118: in Kwaiatek A. et al (2014, 2015)
Please, correct all mistakes: Check all the names of the bacteria, they have to be written in italics.
Please, add a conclusions section.
Author Response
Thank you for all comments. Undoubtedly, these comments have improved the results and the text clarity.
Authors should specify what type of bacterial strain is Neisseria gonorrhoeae FA1090
Our response: Neisseria gonorrhoeae FA1090 (ATCC® 700825™) was originally isolated from the endocervix of a woman with probable disseminated gonococcal infection in 1981(Nachamkin et al., 1981). The strain is well-characterized, serum-resistant, streptomycin-resistant and proline-requiring. N. gonorrhoeae strain FA1090 is extensively used in studies of gonococcal pathogenesis, as it is the first gonococcal strain for which the entire genome sequence was determined (NC_002946.2).
Nachamkin, J. G. Cannon, R. S. Mittler, Monoclonal antibodies against Neisseria gonorrhoeae: production of antibodies directed against a strain-specific cell surface antigen. Infection and immunity 32, 641-648 (1981).
We have completed the text. Please, refer lines 73-78.
Delete these paragraphs, please, put only the reference in parentheses: Page 2, line 77: in Dillard J.P. (2011) Page 3, line 103: in Adamczyk-Poplawska M. et al (2018) Page 3, line 118: in Kwiatek A. et al (2014, 2015)
Our response: We have deleted these paragraphs and included proposed references.
Please, correct all mistakes: Check all the names of the bacteria, they have to be written in italics.
Our response: The manuscript was edited and corrected.
Please, add a conclusions section.
Our response: We have extended a conclusions section. Please refer lines 700-704.
The manuscript needs to be extensively edited for grammar/English, some examples include:
Our response: The manuscript was edited for grammar/English. We have obtained the certificate form AJE (https://www.aje.com) (verification code: 3716-99FB-D00E-72A8-A1CD).
Please note that the numeration and legends of figures have been changed (both in non-supplementary and supplementary).